# Low prevalence of bloodstream infection and high blood culture contamination rates in patients with COVID-19

David Yu[1,2¤], Karolina Ininbergs[1,3], Karolina Hedman[3], Christian G. Giske[1,3], Kristoffer Strålin[4,5], Volkan Özenci[1,3¤]*

1 Division of Clinical Microbiology, Department of Laboratory Medicine, Karolinska Institutet, Stockholm, Sweden, 2 Functional Area of Emergency Medicine, Karolinska University Hospital, Stockholm, Sweden, 3 Department of Clinical Microbiology, Karolinska University Hospital, Stockholm, Sweden, 4 Department of Medicine Huddinge, Karolinska Institutet, Stockholm, Sweden, 5 Department of Infectious Diseases, Karolinska University Hospital, Stockholm, Sweden

¤ Current address: Division of Clinical Microbiology F 72, Karolinska Institutet, Karolinska University Hospital, Stockholm, Sweden

* volkan.ozenci@ki.se

**Data Availability Statement:** All relevant data are within the manuscript and its Supporting Information files.

## Abstract

### Purpose

In the management of COVID-19, knowledge is lacking on the frequency of secondary bacterial infections and on how empirical antibiotic therapy should be used. In the present study, we aimed to compare blood culture (BC) results of a COVID-19 patient cohort with two cohorts of patients without detected COVID-19.

### Methods

Using a retrospective cohort study design of patients subjected to BC in six tertiary care hospitals, SARS-CoV-2 positive patients from March 1 to April 30 in 2020 (COVID-19 group) were compared to patients without confirmed SARS-CoV-2 during the same period (control group-2020) and with patients sampled March 1 to April 30 in 2019 (control group-2019). The outcomes studied were proportion of BC positivity, clinically relevant growth, and contaminant growth.

### Results

In total 15,103 patients and 17,865 BC episodes were studied. Clinically relevant growth was detected in 197/3,027 (6.5%) BC episodes in the COVID-19 group compared to 717/6,663 (10.8%) in control group-2020 (p<0.0001) and 850/8,175 (10.4%) in control group-2019 (p<0.0001). Contamination was present in 255/3,027 (8.4%) BC episodes in the COVID-19 group compared to 330/6,663 (5.0%) in control group-2020 (p<0.0001) and 354/8,175 (4.3%) in control group-2019 (p<0.0001).

**Funding:** The authors received no specific funding for this work.

**Competing interests:** The authors have declared that no competing interests exist.

## Conclusion

In COVID-19 patients, the prevalence of bloodstream bacterial infection is low and the contamination rate of BC is high. This knowledge should influence guidelines regarding blood culture sampling and empirical antibiotic therapy in COVID-19 patients.

## Introduction

Secondary bacterial infections are a major clinical problem in patients with influenza virus infections and have previously been reported to be associated with poor disease outcome [1, 2]. Bloodstream infections (BSIs) remain one of the most common and life-threatening complications in patients with severe viral infections. Epidemiological data of secondary BSIs might therefore play a significant role in reducing mortality and morbidity rates due to COVID-19. A retrospective study from USA reported that patients with COVID-19 have low bacteremia rates than controls [3]. The blood culture routines and the characteristics of COVID-19 patients differ significantly between centers and geographic locations. Therefore, there is an imminent need for studies on BSIs in COVID-19 to understand their importance for disease outcome.

The aim of the present study is to analyze blood culture data of a cohort of COVID-19 patients and compare it with two cohorts of patients without COVID-19.

## Materials and methods

### Setting

The study was performed between 1 March 2020 and 30 April 2020 at Karolinska University Hospital, which serves a population of 2,436,767. The Karolinska University Laboratory receives blood culture specimens from six tertiary care hospitals in the greater Stockholm area and surrounding cities and suburbs. Historically, we have a yearly 10% increase in numbers of our blood culture bottles at Karolinska University Laboratory without any change in contamination rates.

### Study design

Blood cultures collected from patients with COVID-19 and controls were analyzed retrospectively. The blood culture data was retrieved from the laboratory information system (wwLab/ADBakt, Autonik AB, Nykoping, Sweden) using QlikView (Qlik, King of Prussia, PA, USA).

### Study population

**Patients with COVID-19.** Patients were considered to have COVID-19 if they were positive for SARS-CoV-2 RNA by reverse transcriptase PCR in respiratory secretions. Blood culture results from the COVID-19 patients registered between 1 March and 30 April in 2020 and were included and referred to as "COVID-19 group" herein after.

**Control groups.** The study included two control groups, i.e. one historical control group with blood culture results registered between 1 March and 30 April in 2019 (referred to as "control group-2019") and one contemporary control group with blood culture results registered between 1 March and 30 April in 2020, and with no confirmed PCR-positivity for SARS-CoV-2 (referred to as "control group-2020"). In the beginning of the pandemic, testing was only done in patients with symptoms consistent with COVID-19 and not all admitted patients. In the present study, the control group-2020 therefore consisted of a mix of negative

patients and patients not tested. However, all patients with COVID-19-like symptoms were tested. Therefore, it is reasonable to assume that the patients not tested for SARS-COV-2 did not have clinical findings of COVID-19.

### Laboratory methods

**Blood cultures.** Three different blood culture bottles were used in the study; BacT/Alert FA Plus aerobic, BacT/ALERT-PF Plus pediatric and BacT/Alert FN Plus anaerobic plus bottles. Bottles were incubated in BacT/ALERT Virtuo (bioMérieux, Durham, NC, USA) blood culture system until they signaled positive or for a maximum of five days.

The Gram stains were done directly from positive blood culture bottles. According to the result from the staining, specimen from the positive bottles were subcultured onto relevant agar plates. The microorganisms grown on the agar plates were identified by Bruker MALDI-TOF MS. Antimicrobial susceptibility testing was performed by disc diffusion and the results were interpreted following EUCAST recommendations (www.eucast.org).

**SARS-CoV-2 RT-PCR.** Testing for SARS-CoV-2 was performed using three different RT-PCR assays: cobas SARS-CoV-2 (Roche Molecular Systems, Inc., Branchburg, NJ), Xpert Xpress SARS-CoV-2 (XPRSARS-COV2-10) (Cepheid, Sunnyvale, CA) or an in-house developed assay based on Corman et al. [4] targeting the E- and RdRP-genes, with modifications of primers according to Edén A et al. (Neurology, in revision); (Supplementary methods).

### Data analysis

The data on blood cultures were presented as individual BSI episodes, from here on called only "episode". More than one episode from the same patient could be included, however, to be defined as a new episode a minimum of 72 h had to pass between sampling of the same patient. In case of more than 4 bottles taken in a single episode, only the first four bottles were considered in the analysis. Following isolates were considered as contaminants if they grew in less than 3 out of 4 blood culture bottles: *Bacillus* spp., *Corynebacterium* spp., *Cutibacterium* spp., coagulase negative staphylococci (CoNS), *Micrococcus* spp., *Cellulomonas* spp., *Lactobacillus* spp., *Dermabacter* spp., *Facklamia* spp., *Rothia* spp., *Exiguobacterium* spp., *Brevibacterium* spp., and *Trueperella* spp.

### Statistical analysis

The statistical analyses were performed with GraphPad Prism 5.0 (GraphPad Software, San Diego, CA). The blood culture results in patients with COVID-19, control group-2020 and control group-2019 were compared using the Pearson's chi-square test. Values of P $<0.05$ were considered as statistically significant.

## Results

In total, 58,704 blood culture bottles from 17,865 episodes in 15,103 patients were studied. The study flow chart is depicted in Fig 1. The patients of the study groups had the following characteristics; COVID-19 group, 790/2,240 (35.3%) female, mean (Standard Deviation [SD]) age 64 (18) years; control group-2020, 2,789/6,022 (46.3%) female, mean (SD) age 57 (26) years; and control group-2019, 3,322/6,841 (48.6%) female, mean (SD) age 60 (26) years.

### Overall blood culture positivity

The COVID-19 group consisted of 2240/8262 (27%) of all patients sampled for blood cultures during the study period in 2020. The total number of blood cultured patients during the 2020

Total N = 15103 patients

Included periods:
SARS-COV-2: 1 March - 30 April 2020
Control group 2020: 1 March - 30 April 2020
Control group 2019: 1 March - 30 April 2019

| SARS-CoV-2 group: | Control group 2020: | Control group 2019: |
|---|---|---|
| 2240 patients | 6022 patients | 6841 patients |
| 3027 episodes | 6663 episodes | 8175 episodes |

**Fig 1. Flow chart of the study population.**

study period, 8262, is an increase of 1421 (20%), compared to the same period last year (total 6841 patients). In 511/2,240 (22.8%) patients in the COVID-19 group, there were two or more episodes during the study period. In the control groups, BCs were obtained from 6022 and 6841 patients in control group-2020 and control group-2019, respectively. In control group-2020, 459 (7.6%) patients had two or more episodes. In control group-2019, 910 (13.3%) patients had two or more episodes. In total 3,027 episodes in the COVID-19 group, 6,663 in control group-2020 and 8,175 in control group-2019 were studied. Considering episodes, growth of microorganisms in BC was detected in 433/3,027 (14.3%) episodes in the COVID-19 group, compared with 1,015/6,663 (15.2%) in control group-2020 (non-significant) and 1,153/8,175 (14.1%) in control group-2019 (non-significant) (Table 1).

**Table 1. Bloodstream infection episode data for patients with COVID-19 and both control groups.**

| Episode type | COVID-19 | Control group-2020 | Control group-2019 |
|---|---|---|---|
| **Included episodes, N** | **3027** | **6663** | **8175** |
| Episodes with growth, n (%) | 433 (14.3) | 1015 (15.2) | 1153 (14.1) |
| Episodes with clinically relevant growth, n (%) | 197 (6.5) | 717 (10.8) | 851 (10.4) |
| • Gram positive* | 116 (3.8) | 344 (5.2) | 420 (5.1) |
| • Gram negative* | 64 (1.7) | 306 (4.6) | 351 (4.3) |
| • Yeast* | 2 (0.07) | 11 (0.17) | 8 (0.10) |
| • Polymicrobial episodes** | 27 (0.89) | 56 (0.84) | 72 (0.88) |
| Episodes with contaminant growth, n (%) | 255 (8.4) | 330 (5.0) | 354 (4.3) |
| • Only contaminant growth | 236 (7.8) | 298 (4.5) | 302 (3.7) |
| • Both contaminant and clinically relevant growth | 19 (0.63) | 32 (0.48) | 52 (0.64) |

*Monomicrobial episodes.

**Polymicrobial episode is defined as an episode with occurrence of more than one clinically relevant isolate.

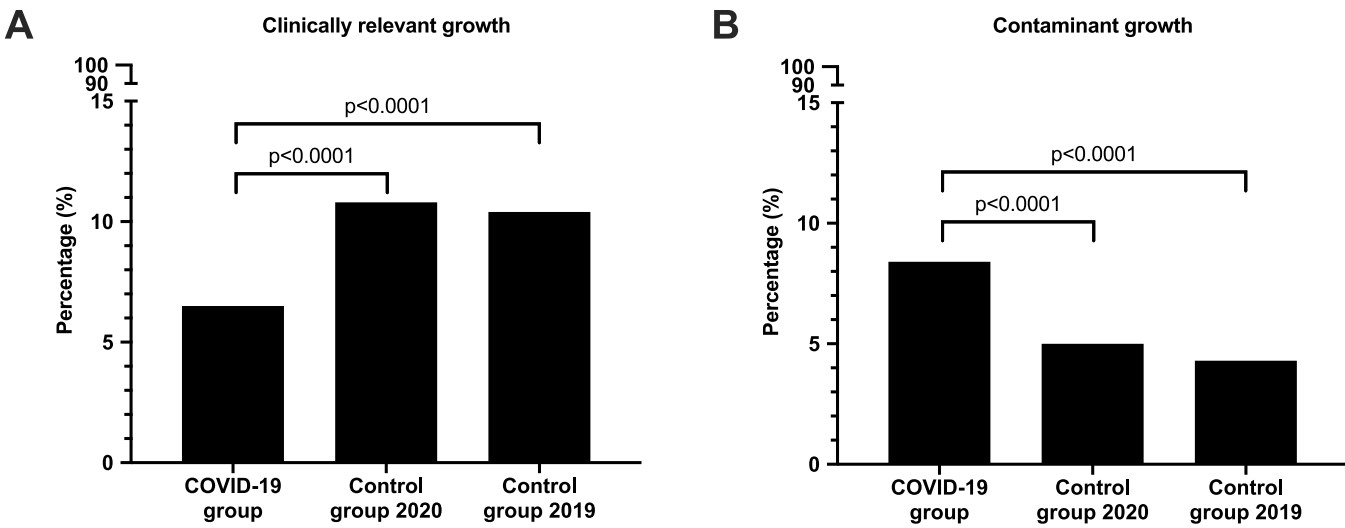

**Fig 2.** Blood culture episodes with clinically relevant growth (Panel A) and with contaminant growth (Panel B). Total number of episodes included for analysis were COVID-19 group: 3,027, Control group-2020: 6,663, Control group-2019: 8,175.

## Clinically relevant growth

Clinically relevant growth was detected in 197/3,027 (6.5%) of episodes in the COVID-19 group, compared with 717/6,663 (10.8%) in control group-2020 (p<0.0001) and 851/8,175 (10.4%) in control group-2019 (p<0.0001) (Table 1, Fig 2 [Panel A]).

When blood cultures with polymicrobial bacteremia were analyzed there was no difference among the three groups studied (Table 1).

## Contaminant growth

Contamination in blood cultures were detected in 255/3,027 (8.4%) episodes in patients with COVID-19, compared with 330/6,663 (4.95%) episodes in control group-2020 (p<0.0001) and 354/8,175 (4.33%) episodes in control group-2019 (p<0.0001). The two control groups had similar numbers of episodes with contaminant growth (non-significant) (Table 1, Fig 2 [Panel B]).

When relationship between the frequency of contaminants and the hospital localization was analyzed, control group-2019 had higher contamination rates in ICUs than in emergency departments and other clinics. In contrast, the contamination rates were similar in all hospital locations for control group-2020. In the COVID-19 group, contamination rates were higher in all hospital locations compared to the control groups, but more so in the emergency departments and ICUs (Table 2).

**Table 2. Numbers and proportions of contamination in blood cultures from different hospital locations.**

| | COVID-19 | | Control group 2020 | | Control group 2019 | |
|---|---|---|---|---|---|---|
| Hospital location | Contaminant (% of total) | Total episodes (n) | Contaminant (% of total) | Total episodes (n) | Contaminant (% of total) | Total episodes (n) |
| Emergency department | 112 (9.2) | 1221 | 130 (4.8) | 2705 | 110 (4.0) | 2769 |
| Intensive care unit | 70 (10.6) | 659 | 18 (5.6) | 319 | 16 (7.9) | 202 |
| Other hospital locations | 73 (6.4) | 1147 | 182 (5.0) | 3639 | 228 (4.4) | 5204 |

**Table 3. Distribution of microorganisms isolated from blood cultures in patients with COVID-19 and both control groups.**

|  | COVID-19 | Control group-2020 | Control group-2019 |
|---|---|---|---|
| **All isolates, N** | 226 | 781 | 940 |
| **Gram-positive bacteria\* n (%)** | 150 (66) | 385 (49) | 487 (52) |
| • Coagulase negative staphylococci\*\* | 49 (22) | 55 (7.1) | 58 (6.2) |
| • *Staphylococcus aureus* | 44 (19) | 124 (16) | 162 (17) |
| • *Enterococcus* spp. | 23 (10) | 46 (5.9) | 75 (8.0) |
| • Viridans group streptococci | 22 (9.7) | 64 (8.2) | 63 (6.7) |
| • Beta-hemolytic streptococci | 6 (2.7) | 47 (6.0) | 50 (5.3) |
| • *Streptococcus pneumoniae* | 3 (1.3) | 21 (1.4) | 51 (1.6) |
| • Other Gram-positive bacteria† | 3 (1.3) | 28 (3.6) | 28 (3.0) |
| **Gram-negative bacteria\* n (%)** | 66 (29) | 352 (45) | 397 (42) |
| • *Escherichia coli* | 34 (15) | 213 (27) | 242 (26) |
| • Other Enterobacterales | 24 (11) | 111 (14) | 117 (13) |
| • *Pseudomonas aeruginosa* | 7 (3.1) | 13 (1.7) | 14 (1.5) |
| • Other Gram-negative bacteria† | 1 (0.4) | 15 (1.9) | 24 (2.6) |
| **Anaerobic bacteria n (%)†** | 7 (3.1) | 32 (4.1) | 40 (4.3) |
| **Yeast n (%)†** | 3 (1.3) | 11 (1.4) | 15 (1.6) |

\*Not including anaerobic bacteria

\*\*Including *Staphylococcus epidermidis*

†Other bacteria and yeast are shown in S1 Table.

Similar results were observed when numbers of bottles with contaminants were analyzed. In total 337/10,504 (3.2%) bottles in the COVID-19 group were contaminated, compared to 433/21,261 (2.0%) in control group-2020 (p<0.0001) and 470/26,939 (1.7%) in control group-2019 (p<0.0001).

## Diversity of microorganisms in blood cultures

There was a significant diversity in microorganisms detected from blood cultures among the three groups studied. Gram-positive growth was significantly higher in patients with COVID-19, 150/226 (66%) isolates, than in control group-2020 and -2019, 385/781 (49%) 487/940 (52%) isolates respectively (p<0.0001 for both comparisons) (Table 3). The two control groups had similar levels of numbers of Gram-positive growth (non-significant). In contrast, Gram-negative isolates was significantly fewer in patients with COVID-19, 66/226 (29%) isolates, than in control group-2020 353/781 (45%) isolates and -2019 398/940 (42%) isolates (p<0.0001 and p<0.001 respectively). The two control groups had similar levels of numbers of Gram-negative isolates (non-significant).

The most common three microorganisms detected in blood cultures from COVID-19 patients were CoNS, *Staphylococcus aureus* and *Escherichia coli*. In contrast, for both control groups the three most common isolates were *E. coli*, *S. aureus* and other *Enterobacterales*.

Patients with COVID-19 had higher numbers of episodes with clinically relevant CoNS than in control group-2020 and -2019 (p<0.0001 for both comparisons). There was no difference in numbers of episodes with CoNS between the two control groups (non-significant). In contrast, lower numbers of *Escherichia coli* were observed in patients with COVID-19 (15%) compared to control group-2020 (27%) and -2019 (26%) (p = 0.0001 and p = 0.0005, respectively). There was no significant difference for *S. aureus*, anaerobes and yeasts among the three groups studied. A detailed list of all microorganisms can be found in supplements (S1 Table).

**Table 4. Time to detection of microorganisms in positive blood cultures in patients with COVID-19 and both control groups.**

| Time to detection | COVID-19 | Control group | Control group |
|---|---|---|---|
| | (n = 820) | 2020 (n = 2185) | 2019 (n = 2663) |
| Mean (SD [h]) | 22.8 (17.6) | 18.4 (15.3) | 18.2 (17.2) |
| Median (IQR [h]) | 18.5 (11.5) | 14.0 (10.7) | 13.2 (10.4) |

SD: standard deviation. IQR: interquartile range.

### Time to detection in blood cultures

The time to detection (TTD) in positive blood culture bottles differed between the three groups analyzed. The mean (SD) TTD was 22.8 (17.6) in COVID-19 group whereas control group-2020 and control group-2019 had 18.4 (15.3) and 18.2 (17.2) h (Table 4).

When the incubation period for blood cultures were analyzed in a total of 5568 bottles, we observed that the vast majority (93–96%) of the bottles signaled positive in 48 h. An additional 4–5% signaled positive in 4 days. The remaining BC that signaled positive in 5 days were 2% in COVID-19 group and 1% in both control groups (Table 5).

### Discussion

Blood culture is the gold standard for detection of microorganisms in patients with BSI. We presented the blood culture findings in patients with COVID-19 and in other patients with suspected BSIs from a contemporary and a historical control group.

In the present study the overall blood culture positivity rate was similar in all three groups analyzed. However, the proportion of episodes with clinically relevant growth was significantly lower in in patients with COVID-19 than both control groups. Although in our study the true incidence of bacteremia was not known, the proportion of episodes with clinically relevant growth correlates with previous data on clinical characteristics in COVID-19, where bacteremia was observed in 5.6% of cases [5] and septic shock in 4% of cases [6]. The reason for lower bacteremia rates in patient with COVID-19 is largely unknown. Patients with severe COVID-19 fulfill the sepsis-3 criteria for sepsis [7] and the term viral sepsis has been introduced [8]. In patients who are hospitalized for COVID-19, it is thus difficult to use clinical and laboratory parameters to differentiate between the viral component and a potential bacterial component. The low rate of relevant bacteremia indicates that the viral component is predominant in COVID-19. There was a higher proportion of patients with more than one suspected BSI episode in the COVID-19 group compared to the control groups. The reason for this is not known.

In a recent study, from New York, USA, it is reported that only 3.8% of COVID-19 patients had positive blood cultures which was significantly lower than the controls in that study [3]. The present data differs from the NY study since the overall BC positivity rate in our COVID-

**Table 5. Proportions of positive blood cultures signaling positive during given time intervals in the blood culture system.**

| Time to detection | COVID-19 | Control group 2020 | Control group 2019 |
|---|---|---|---|
| | (n = 820) | (n = 2185) | (n = 2663) |
| Day 1–2 | 93% | 96% | 95% |
| Day 3 | 4% | 3% | 3% |
| Day 4 | 1% | 1% | 1% |
| Day 5 | 2% | 1% | 1% |

19 group was 14.3% and did not differ from controls. Moreover, clinically relevant growth in the present COVID-19 cohort was 6.5% in comparison to 1.8% in COVID-19 patients reported by Sepulveda et al. [3]. The underlying reason for these differences is unknown. It is reasonable to assume that the blood culture routines and the characteristics of COVID-19 patients were different. However, both studies showed that the clinically relevant growth is lower in patients with COVID-19 than in controls.

The present results show that bacterial and fungal BSIs are uncommon in patients with COVID-19 and it is warranted to establish stringent clinical criteria for empiric antibiotic treatment for BSI in these patients.

The species composition of microorganisms isolated from blood cultures from patients with COVID-19 and controls differed. When the three most common microorganisms isolated from the positive bottles were considered, patients with COVID-19 had significantly higher rates of clinically relevant CoNS than both control groups (p<0.0001 for both comparisons). In contrast, both control groups had higher rates of *E. coli* compared to COVID-19 patients (p = 0.0001 and p = 0.0005, respectively). All three groups had similar levels of *S. aureus* (non-significant). The underlying differences in occurrence of BSI with CoNS and *E. coli* in patients with COVID-19 and controls might be important in empiric antibiotic treatment of these patients. Overall, our results emphasize the importance of antimicrobial stewardship in the treatment of COVID-19 patient to minimize the threat of superinfections [9].

Contamination is a major problem in blood cultures. It was recently reported that COVID-19 patients had higher proportion of cultures that likely represented contamination with normal skin microbiota than controls [3]. However, the study did not analyze the growth of contaminants further. Growth of normal skin microbiota might be clinically relevant. The present study analyzed the contaminants in detail both at episode and BC bottle level, by using an algorithm to discriminate possible clinically relevant growth of normal skin flora bacteria [10, 11]. Patients with COVID-19 had 3.2% blood culture bottles (8.4% episodes) contaminated as compared to 2.04% (5.0% episodes) and 1.74% (4.2% episodes) in the two control groups, respectively. Under normal circumstances, blood cultures received in our center has low contamination rates as shown in the two control groups studied. In contrast, the blood culture bottles in the COVID-19 group exceeded the recommended rate of contamination of <3% according to CLSI guidelines [12], which may lead to unnecessary antibiotic use and longer hospital stay for these patients. The underlying reason for high contamination rates in patients with COVID-19 is not known. In the present study, contamination rates were higher in emergency departments and ICUs compared to other units in the COVID-19 group. The relationship between high stress environments and BC contamination rates has been previously reported [13, 14]. The patient characteristics and the work pace in emergency departments and ICUs differs from the other units. It is reasonable to suggest that the stressful working environment in these two units with well-known risk to be exposed to SARS-CoV-2 might play an important role in higher contamination rates observed in COVID-19 group.

TTD of positive blood cultures might be a relevant parameter in comparing different patient populations [15]. In the present study we showed that the mean TTD in COVID-19 was approximately 20% longer compared to controls. It is reasonable to suggest that the longer TTD in COVID-19 group is based on higher rates of contaminants in this group.

The recommended incubation period for blood cultures is 5 days. In line with a recently published study, we observed that 98–99% of the BC bottles in all three groups signaled positive in 4 days [3]. The present data support the assumption that BC can be incubated for a maximum of 4 days when it is necessary.

To our knowledge, this is the first European study analyzing blood culture data from patients with COVID-19 and has several strengths. The inclusion of over 15,000 patients from

six tertiary care hospital is an important strength of the study design. The study analyzed the clinically relevant growth and contaminants in detail and had relatively high positive blood culture rates in all three groups studied.

The present study also has important limitations. First, we did not have access to baseline clinical data such as comorbidity, disease duration, length of hospital stay and treatment of patients in the three study groups. Therefore, assessment of the impact of differences in patient characteristics on the BC results could not be analyzed. Second, we did not include a control group with another viral respiratory infection, such as influenza, during the same season. Third, as the present study focused on BSI in COVID-19, data regarding other culture results were not analyzed, which precluded the analysis of other secondary infections such as pneumonia.

## Conclusions

The present study shows that patients with COVID-19 have low prevalence of BSI and a higher rate of contamination in blood cultures. Further clinical studies are warranted in order to improve blood culture-based diagnostics in patients with COVID-19.

## Supporting information

**S1 Table. Distribution of all microorganisms isolated from blood cultures.**
(DOCX)

## Author Contributions

**Data curation:** David Yu, Karolina Ininbergs, Karolina Hedman.

**Formal analysis:** David Yu, Karolina Ininbergs.

**Methodology:** David Yu, Christian G. Giske, Kristoffer Strålin, Volkan Özenci.

**Project administration:** Volkan Özenci.

**Supervision:** Volkan Özenci.

**Writing – original draft:** David Yu, Christian G. Giske, Volkan Özenci.

**Writing – review & editing:** Volkan Özenci.

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
