## [Decision Letter · Decision Letter 0]

21 Aug 2020

PONE-D-20-20448

Low prevalence of bloodstream infection and high blood culture contamination rates in patients with COVID-19

PLOS ONE

Dear Dr. Özenci,

Thank you for submitting your manuscript to PLOS ONE. After careful consideration, we feel that it has merit but does not fully meet PLOS ONE’s publication criteria as it currently stands. Therefore, we invite you to submit a revised version of the manuscript that addresses the points raised during the review process.

My evaluation of this manuscript is noted below under "Additional Editor Comments". Please be sure to address these in addition to the comments provided by the reviewers. I have highlighted key points from reviewers in my evaluation as well.

We look forward to receiving your revised manuscript.

Kind regards,

Surbhi Leekha

Academic Editor

PLOS ONE

Additional Editor Comments:

1. In the absence of clinical data, the length of the manuscript should be shortened and the lack of clinical data should be acknowledged as an important limitation.

2. Include numbers and proportions of patients admitted during the study time-frame (in 2020) that underwent SARS-CoV-2 testing. Also state whether testing for SARS-CoV-2 was done on all admitted patients or only those who presented with symptoms, to help understand if the control group represented SARS-CoV2 negative patients, or a mix of tested-negative and patients not tested. If possible, briefly describe the COVID-19 care model and infection control practice in the study period. This would help better understand the context for higher contamination rates.

3. Please include the blood culturing frequency (relative to the number of patients or patient-days) for the three cohorts. The number of contaminated blood cultures may also be related to more blood cultures being obtained.

4. Related to the point above, please be careful with the use of terms “positivity” and “bacteremia rate” both when describing your own results and when comparing with the published literature (e.g., New York data). I believe that in this context, positivity is the positive blood cultures/total number of episodes whereas bacteremia rates are positive blood cultures/total patients. These terms should be described and used consistently.

5. Both reviewers have commented on the lack of relevant clinical data. While including clinical data might be out of the scope of your study, are you able to include data on two important variables: 1) hospital location, and 2) time from admission as relevant clinical variables that would provide important information. Specifically, blood cultures obtained in the ED setting are well described to have high rates of contamination so location where blood cultures were obtained particularly ED vs inpatient should be reported. Second, as pointed out be reviewer 2, it would be clinically useful to compare positivity for contaminants and non-contaminants by time since admission – you could report this stratified by within one week from admission, and after one week from admission.

6. Was any statistical testing done for time to positivity, and proportion positive by time period?

2. Please include the date(s) on which you accessed the laboratory information system to obtain the data used in your study.

3. PLOS ONE requires experimental methods to be described in enough detail to allow suitably skilled investigators to fully replicate and evaluate your study.

See https://journals.plos.org/plosone/s/submission-guidelines#loc-materials-and-methods for more information. To comply with PLOS ONE submission guidelines, in your Methods section, please provide a more detailed description of microorganism identification and antimicrobial susceptibility methodology.

5. Please include your tables as part of your main manuscript and remove the individual files.

Please note that supplementary tables should be uploaded as separate "supporting information" files.

Reviewers' comments:

Reviewer's Responses to Questions

**Comments to the Author**

1. Is the manuscript technically sound, and do the data support the conclusions?

Reviewer #1: Yes

Reviewer #2: Yes

2. Has the statistical analysis been performed appropriately and rigorously? 

Reviewer #1: Yes

Reviewer #2: Yes

3. Have the authors made all data underlying the findings in their manuscript fully available?

Reviewer #1: Yes

Reviewer #2: Yes

4. Is the manuscript presented in an intelligible fashion and written in standard English?

Reviewer #1: Yes

Reviewer #2: Yes

5. Review Comments to the Author

Reviewer #1: a bit lengthy and wordy for the data being presented, and could be presented much more concisely

while writing is grammatically accurate, the English is a little clunky

clinical data would be useful given some of the demographic discrepancies (greater M > F ratio in COVID group)

would favor medians instead of means

would be consistent with abbreviations, especially with bacteria names

line 181 - Enterobacterales isn't a species

might be helpful to note numbers of repeat episodes

any difference in pathogens isolated by hospital day? ie were contaminants more likely to be isolated on admission vs later in the hospitalization which may have represented a clinically relevant pathogen?

Reviewer #2: Thank you for the opportunity to review this interesting and well written manuscript on secondary bacterial BSI in COVD-19 patients. This study adds important information to the literature by providing comparison to non-COVID patients both during the pandemic and historically. In my opinion, this is an important comparison that has not yet been made properly and begins to address some of the missing gaps in knowledge surrounding COVID-19 and bacterial / fungal infections. One area that could be stronger in the discussion is the nature of secondary infection in COVID-19 and an exploration of whether it is directly attributable to COVID-19 or a consequence of increased healthcare exposure and pressures on services.

Abstract

No comments

Introduction

1. Secondary bacterial infections in influenza often refers to respiratory tract infection. The authors of this study have focused on blood culture results. This makes comparison between influenza and COVID-19 in this case challenging.

Method:

1. Were all patients admitted during March and April tested for SARS-CoV-2? If patients with high clinical suspicion were not routinely tested for example this may have led to overlap in the SARS-CoV-2 and negative control group. Similarly, false negatives / positive results may have occurred. This should be acknowledged as a limitation of the study.

2. How were contaminant organisms determined?

3. Were control groups matched in any way? If not how were the control samples selected?

Results:

1. What was the total number of COVID-19 positive patients from which the blood culture pool were selected during the study time period?

2. Doe the authors have data to describe the setting where blood cultures were taken for these patients? For example, were line associated cultures more common in COVID patients? Were more cultures performed in critical care?

3. Can the authors provide any information of antimicrobial prescribing trends during the time periods described?

Discussion:

1. An important point is the reason for contamination of blood cultures. This should be explored further. Is this the nature of PPE use, reduced hand hygiene, or due to staffing issues during expanded critical care capacity?

2. Furthermore, could the shift in observed species be due to reductions in elective intra-abdominal and urinary operations / procedures being performed during this period?

3. It would also be interesting to known whether the absolute number of patients with COVID in the healthcare system during this time. For example, could it be that Gram-negative infections were missed as fever was put down to COVID and less cultures were actually performed?

4. Another important limitation of this study that should be acknowledged and discussed is that the authors have not looked at other microbiological cultures. For example, a major concern during COVID-19 has been the reporting of bacterial/fungal respiratory infection. The authors are unable to comment on secondary infections such as HAP and VAP given their focus on blood cultures only.

5. Furthermore, by not providing clinical details, it is not possible to know whether patients with and without COVID-19 are matched or different cohorts (e.g. number of critically unwell patients, number of line days per patient, number of patients with respiratory versus GI pathology etc.).

6. One area that could come through stronger in the discussion is the nature of secondary infection in COVID-19 and exploration of whether it is directly attributable to COVID-19 or a consequence of increased healthcare exposure and pressures on services. For example, are these infections similar to routine HCAI’s associated with ICU / increased line days? And are we seeing more of them due to expanded ICU capacity during this period?

6. PLOS authors have the option to publish the peer review history of their article (what does this mean?). If published, this will include your full peer review and any attached files.

Reviewer #1: No

Reviewer #2: No

---

## [Author Response · Author response to Decision Letter 0]

17 Sep 2020

Reply to the Editor and Reviewers’ Comments

Additional Editor Comments:

1. In the absence of clinical data, the length of the manuscript should be shortened and the lack of clinical data should be acknowledged as an important limitation.

As the Editor suggested the original version of the manuscript is shortened and the limitation with the lack of clinical data is included in the discussion section as follows: “The present study has important limitations. First, we did not have access to base-line clinical data such as co-morbidity, disease duration and treatment of patient in all three groups. Therefore, assessment of the impact of differences in patient characteristics on the blood culture results could not be analyzed….”

2. Include numbers and proportions of patients admitted during the study time-frame (in 2020) that underwent SARS-CoV-2 testing. Also state whether testing for SARS-CoV-2 was done on all admitted patients or only those who presented with symptoms, to help understand if the control group represented SARS-CoV2 negative patients, or a mix of tested-negative and patients not tested. If possible, briefly describe the COVID-19 care model and infection control practice in the study period. This would help better understand the context for higher contamination rates.

The contemporary group in 2020 was heterogenous in regard to SARS-COV-2 testing. The number of patients admitted to the hospital during the study periods is not known to us, and therefore the proportion of tested patients is unknown. The testing routine for SARS-COV-2 varied during the study period. In the beginning of the pandemic, testing was only done in patients with symptoms consistent with COVID-19 and not all admitted patients. In the present study, the control group therefore consisted of a mix of negative patients and patients not tested. However, all patients with COVID-19-like symtoms were tested. Therefore, it is reasonable to assume that the patients not tested for SARS-COV-2 did not have clinical findings of COVID-19. 

The question on the COVID-19 care model and infection control practice in the study period is very relevant. However, the care of the COVID-19 patients was not consistent during the early phase of the pandemic in Sweden as in other countries and it is difficult to describe shortly in the current manuscript. We believe it is also out of the scope of the current study. 

3. Please include the blood culturing frequency (relative to the number of patients or patient-days) for the three cohorts. The number of contaminated blood cultures may also be related to more blood cultures being obtained.

The study included patient samples from six tertiary care hospitals. Therefore, it is rather difficult to obtain exact numbers of patients treated during this period. 

The COVID-19 patients were in total 2240 (27%), of all patients (8262) sampled for blood cultures during the study period. The total number of blood cultured patient during the study period, 8262, is an increase of 1421 (20%), compared to the same period last year (total 6841 patients). Normally we have a yearly 10% increase in numbers of our blood culture bottles at Karolinska without any change in contamination rates. Therefore it is highly unlikely that the increase in number of contaminated blood cultures is related to more blood cultures being obtained. 

4. Related to the point above, please be careful with the use of terms “positivity” and “bacteremia rate” both when describing your own results and when comparing with the published literature (e.g., New York data). I believe that in this context, positivity is the positive blood cultures/total number of episodes whereas bacteremia rates are positive blood cultures/total patients. These terms should be described and used consistently.

In the present study, a previous established algorithm was used to determine isolates as clinically relevant or contamination. Therefore, we used “positivity” to denote overall positive blood culture episodes, where as “clinically relevant” is used to describe isolates which represent true bacteremia. As the analyses in our study is based on episodes, we avoid the term bacteremia and instead use “positive episodes” and “episodes with clinically relevant growth” in the revised version.

5. Both reviewers have commented on the lack of relevant clinical data. While including clinical data might be out of the scope of your study, are you able to include data on two important variables: 1) hospital location, and 2) time from admission as relevant clinical variables that would provide important information. Specifically, blood cultures obtained in the ED setting are well described to have high rates of contamination so location where blood cultures were obtained particularly ED vs inpatient should be reported. Second, as pointed out be reviewer 2, it would be clinically useful to compare positivity for contaminants and non-contaminants by time since admission – you could report this stratified by within one week from admission, and after one week from admission.

We acknowledge the lack of clinical data. As the reviewers suggested, the data on blood culture results in relation to hospital locations is included and the data on the following three groups are presented in the revised version: Emergency department, intensive care unit, and others. The data is presented in a new table (Table 3) as follows:

Table 3. Positive episodes and proportion of contamination in blood cultures from different hospital locations.

Hospital location COVID-19 Control group 2020 Control group 2019

 Positive episodes Contaminant (% of positive) Positive episodes Contaminant (% of positive) Positive episodes Contaminant (% of positive)

Emergency department 183 112 (61.2) 415 130 (31.3) 352 110 (31.2)

Intensive care unit 108 70 (64.8) 32 18 (56.2) 30 16 (53.3)

Other hospital locations 142 73 (58.9) 568 182 (32.0) 771 228 (29.6)

The information on time since admission is unfortunately not available, as we did not have access to the patient records including time that the patient was admitted to the hospital. We had only the time of BC sampling as it is available in our laboratory information system. 

6. Was any statistical testing done for time to positivity, and proportion positive by time period?

No statistical testing was done for time to positivity and proportion positive by time period. 

We followed the PLOS ONE's style requirements. 

2. Please include the date(s) on which you accessed the laboratory information system to obtain the data used in your study.

All data was retrieved from the LIS on 30th of April 2020.

3. PLOS ONE requires experimental methods to be described in enough detail to allow suitably skilled investigators to fully replicate and evaluate your study.

The study is based on data collection from the LIS. The study can easily be reproduced by a similar search in any LIS including ours.

See https://journals.plos.org/plosone/s/submission-guidelines#loc-materials-and-methods for more information. To comply with PLOS ONE submission guidelines, in your Methods section, please provide a more detailed description of microorganism identification and antimicrobial susceptibility methodology.

As the data that were not shown was not a part of the core results, we have decided to remove it from the revised version of the manuscript.

5. Please include your tables as part of your main manuscript and remove the individual files.

Please note that supplementary tables should be uploaded as separate "supporting information" files.

Thank you, we will submit the tables and supplementary tables as suggested. 

Reply to Reviewers’ comments

Comments to the Author

1. Is the manuscript technically sound, and do the data support the conclusions?

Reviewer #1: Yes

Reviewer #2: Yes

2. Has the statistical analysis been performed appropriately and rigorously? 

Reviewer #1: Yes

Reviewer #2: Yes

3. Have the authors made all data underlying the findings in their manuscript fully available?

Reviewer #1: Yes

Reviewer #2: Yes

4. Is the manuscript presented in an intelligible fashion and written in standard English?

Reviewer #1: Yes

Reviewer #2: Yes

5. Review Comments to the Author

Reviewer #1: a bit lengthy and wordy for the data being presented, and could be presented much more concisely while writing is grammatically accurate, the English is a little clunky clinical data would be useful given some of the demographic discrepancies (greater M > F ratio in COVID group) would favor medians instead of means would be consistent with abbreviations, especially with bacteria names

As the Reviewer suggested the revised version of the manuscript is shortened and medians were used instead of means.

line 181 - Enterobacterales isn't a species 

Corrected

might be helpful to note numbers of repeat episodes 

Done

any difference in pathogens isolated by hospital day? ie were contaminants more likely to be isolated on admission vs later in the hospitalization which may have represented a clinically relevant pathogen?

Unfortunately, we did not have access to patient journals. The revised manuscript includes data on hospital location when BC sampling was performed. As it is shown in Table 3, both control groups had higher contamination rates in ICU than in emergency departments and other clinics. However, the contamination rates were similar among emergency departments, ICU and other clinics for patients with COVID-19 which indicates that the high rate of contamination in patients with COVID-19 is probably related to difficulties in optimal blood culture sampling with the protective equipment and stress level. 

Reviewer #2: Thank you for the opportunity to review this interesting and well written manuscript on secondary bacterial BSI in COVD-19 patients. This study adds important information to the literature by providing comparison to non-COVID patients both during the pandemic and historically. In my opinion, this is an important comparison that has not yet been made properly and begins to address some of the missing gaps in knowledge surrounding COVID-19 and bacterial / fungal infections.

We thank the Reviewer for acknowledging the importance of our study.

 One area that could be stronger in the discussion is the nature of secondary infection in COVID-19 and an exploration of whether it is directly attributable to COVID-19 or a consequence of increased healthcare exposure and pressures on services.

The question of the underlying cause for lower bacteremia rates and higher contamination rates in COVID-19 is highly relevant. However, the results of the present study are not sufficient to determine the underlying cause as clinical parameters were lacking. 

Abstract

No comments

Introduction

1. Secondary bacterial infections in influenza often refers to respiratory tract infection. The authors of this study have focused on blood culture results. This makes comparison between influenza and COVID-19 in this case challenging.

We agree with the Reviewer that bacterial LRTI are the major secondary bacterial infections after viral RTI. However, the aim of the current study was to analyze the association between COVID-19 and BSI. 

Method:

1. Were all patients admitted during March and April tested for SARS-CoV-2? If patients with high clinical suspicion were not routinely tested for example this may have led to overlap in the SARS-CoV-2 and negative control group. Similarly, false negatives / positive results may have occurred. This should be acknowledged as a limitation of the study.

This relevant question was also raised by the Editor. In the present study, during the 2020 study period, the testing routine for SARS-COV-2 was not consistent. During the pandemic, SARS-COV-2 testing was initially performed only for patients with clinical signs of COVID-19. Therefore, there is a possibility that the negative control group might have included patients that had been positive if a ”test all” strategy had been employed. However, as virtually all patients with clinical signs were tested, it is possible to assume that the control group-2020 represent patients without clinical significant COVID-19.

2. How were contaminant organisms determined?

There is no gold standard definition of contaminants in blood cultures. We used a modified version of a previously published algorithm by our group as well as others (Yu et al 2020; Bekeris et al., 2005; Dawson, 2014).

3. Were control groups matched in any way? If not how were the control samples selected?

Control groups were not matched. Control samples 2020 were taken during the same study period as SARS-COV-2 samples. 

Results:

1. What was the total number of COVID-19 positive patients from which the blood culture pool were selected during the study time period?

The total numbers of COVID-19 patients under the study period were 7617.

2. Doe the authors have data to describe the setting where blood cultures were taken for these patients? For example, were line associated cultures more common in COVID patients? Were more cultures performed in critical care?

Whether sampling from peripheral or central vein catheters were not documented consistently. However, we described the numbers of positive samples from the three hospital locations namely emergency department, intensive care unit, and others in the revised version of the manuscript. 

3. Can the authors provide any information of antimicrobial prescribing trends during the time periods described?

 This is an important question. The study included six tertiary care hospitals and the data on antimicrobial prescribing trends during the time periods was not available since we did not have access to patient journals. 

Discussion:

1. An important point is the reason for contamination of blood cultures. This should be explored further. Is this the nature of PPE use, reduced hand hygiene, or due to staffing issues during expanded critical care capacity?

The underlying reasons for higher contamination rates are probably multifactorial and consist of all of the above-mentioned problems. Detailed analysis of these factors could not be made, and they are out of the scope of the study. 

2. Furthermore, could the shift in observed species be due to reductions in elective intra-abdominal and urinary operations / procedures being performed during this period?

The patient profile in these six hospitals and in almost all hospitals in Sweden and in other countries has indeed changed dramatically as the Reviewer indicated. It is possible that the reduction of elective patients reflects the microorganisms isolated from BC. However, this factor was not analyzed in the current study. 

3. It would also be interesting to known whether the absolute number of patients with COVID in the healthcare system during this time. For example, could it be that Gram-negative infections were missed as fever was put down to COVID and less cultures were actually performed?

The COVID-19 patients were in total 2240/8262 (27%), of all patients sampled for blood cultures during the study period. The total number of blood cultured patients during the study period were 8262 and 20% higher compared to the same period last year (total 6841 patients in 2019). 

The total number of COVID-19 positive patients during the study period was 7617, which includes also non-hospitalized patients with mild symptoms. It is reasonable to assume that the results are influenced by the shift in patient profiles, as mentioned above. However, our data precludes conclusions regarding potential effects from clinical sampling strategies.

4. Another important limitation of this study that should be acknowledged and discussed is that the authors have not looked at other microbiological cultures. For example, a major concern during COVID-19 has been the reporting of bacterial/fungal respiratory infection. The authors are unable to comment on secondary infections such as HAP and VAP given their focus on blood cultures only.

We thank the reviewer for acknowledging this limitation, which have been added to the discussion section in the revised manuscript: “Third, as the present study focused on BSI in COVID-19, data regarding other cultures were not analyzed, which precluded the analysis of other secondary infections such as pneumonia. “ 

5. Furthermore, by not providing clinical details, it is not possible to know whether patients with and without COVID-19 are matched or different cohorts (e.g. number of critically unwell patients, number of line days per patient, number of patients with respiratory versus GI pathology etc.).

The lack of clinical data has been emphasized as an important limitation in the revised manuscript.

6. One area that could come through stronger in the discussion is the nature of secondary infection in COVID-19 and exploration of whether it is directly attributable to COVID-19 or a consequence of increased healthcare exposure and pressures on services. For example, are these infections similar to routine HCAI’s associated with ICU / increased line days? And are we seeing more of them due to expanded ICU capacity during this period?

This is an important point, however the cause of secondary infection in COVID-19 is difficult to elucidate from the parameters included in the present study. Most likely, it is multifactorial and requires an analysis on both a microbiological, clinical, and logistical level which is out of the scope of this study.

6. PLOS authors have the option to publish the peer review history of their article (what does this mean?). If published, this will include your full peer review and any attached files.

Do you want your identity to be public for this peer review? For information about this choice, including consent withdrawal, please see our Privacy Policy.

Reviewer #1: No

Reviewer #2: No

När du skickar e-post till Karolinska Institutet (KI) innebär detta att KI kommer att behandla dina personuppgifter. Här finns information om hur KI behandlar personuppgifter. 

Sending email to Karolinska Institutet (KI) will result in KI processing your personal data. You can read more about KI’s processing of personal data here.

---

## [Editor Report · Decision Letter 1]

30 Sep 2020

PONE-D-20-20448R1

Low prevalence of bloodstream infection and high blood culture contamination rates in patients with COVID-19

PLOS ONE

Dear Dr. Özenci,

Thank you for submitting your manuscript to PLOS ONE. After careful consideration, we feel that it has merit but does not fully meet PLOS ONE’s publication criteria as it currently stands. Therefore, we invite you to submit a revised version of the manuscript that addresses the points raised during the review process.

Please add part of your response to the question about which patient population was tested for SARS-CoV-2 to the manuscript:

“The contemporary group in 2020 was heterogenous in regard to SARS-COV-2 testing. The number of patients admitted to the hospital during the study periods is not known to us, and therefore the proportion of tested patients is unknown. The testing routine for SARS-COV-2 varied during the study period. In the beginning of the pandemic, testing was only done in patients with symptoms consistent with COVID-19 and not all admitted patients. In the present study, the control group therefore consisted of a mix of negative patients and patients not tested. However, all patients with COVID-19-like symtoms were tested. Therefore, it is reasonable to assume that the patients not tested for SARS-COV-2 did not have clinical findings of COVID-19.”

Below is your response to the previous question on blood culturing frequency. Please include part of this response in the results and discussion.

“The study included patient samples from six tertiary care hospitals. Therefore, it is rather difficult to obtain exact numbers of patients treated during this period. The COVID-19 patients were in total 2240 (27%), of all patients (8262) sampled for blood cultures during the study period. The total number of blood cultured patient during the study period, 8262, is an increase of 1421 (20%), compared to the same period last year (total 6841 patients). Normally we have a yearly 10% increase in numbers of our blood culture bottles at Karolinska without any change in contamination rates. Therefore it is highly unlikely that the increase in number of contaminated blood cultures is related to more blood cultures being obtained.

Additionally in the results, you report that: ”BCs were obtained from 2,240 patients with COVID-19. In 511/2,240 (22.8%) patients, there were two or more episodes during the study period. In the control groups, BCs were obtained from 6022 and 6841 patients in control group-2020 and control group-2019, respectively. In control group-2020, 459 (7.6%) patients had two or more episodes. In control group-2019, 910 (13.3%) patients had two or more episodes.”

The above suggests that the frequency of **repeat **culturing was higher in the COVID-19 group. I would add that to the discussion.

Thank you for adding Table 2 and distribution of blood cultures by hospital location. However, the comparison of blood culture contamination rates across locations requires looking at contaminated blood cultures as a proportion of the total blood cultures obtained at that location (and not the total positives). Typically ED contamination rates higher than inpatient locations. You want to assess whether that difference was similar for COVID-19 patients vs controls. Please make changes to Table 2 to reflect that, and add to discussion as appropriate.Once again, in the discussion you are comparing your “positivity” rate to bacteremia incidence in published studies. I would clearly state that while you don’t have true incidence in your study, the positivity rate is similar to incidence of bacteremia reported in other studies.

“In the present study the overall blood culture positivity rate was similar in all three groups analyzed. However, the proportion of episodes with clinically relevant growth was significantly lower in in patients with COVID-19 than both control groups. This correlates with previous data on clinical characteristics in COVID-19, where bacteremia was observed in 5.6% of cases and septic shock in 4% of cases and septic shock in 4% of cases”

We look forward to receiving your revised manuscript.

Kind regards,

Surbhi Leekha

Academic Editor

PLOS ONE

---

## [Author Response · Author response to Decision Letter 1]

8 Oct 2020

PONE-D-20-20448R1

Low prevalence of bloodstream infection and high blood culture contamination rates in patients with COVID-19

PLOS ONE

Dear Dr. Özenci,

Thank you for submitting your manuscript to PLOS ONE. After careful consideration, we feel that it has merit but does not fully meet PLOS ONE’s publication criteria as it currently stands. Therefore, we invite you to submit a revised version of the manuscript that addresses the points raised during the review process.

1. Please add part of your response to the question about which patient population was tested for SARS-CoV-2 to the manuscript:

“The contemporary group in 2020 was heterogenous in regard to SARS-COV-2 testing. The number of patients admitted to the hospital during the study periods is not known to us, and therefore the proportion of tested patients is unknown. The testing routine for SARS-COV-2 varied during the study period. In the beginning of the pandemic, testing was only done in patients with symptoms consistent with COVID-19 and not all admitted patients. In the present study, the control group therefore consisted of a mix of negative patients and patients not tested. However, all patients with COVID-19-like symtoms were tested. Therefore, it is reasonable to assume that the patients not tested for SARS-COV-2 did not have clinical findings of COVID-19.”

In the revised “Materials and methods” section the following information is added as the Editor suggested (L94-99): “In the beginning of the pandemic, testing was only done in patients with symptoms consistent with COVID-19 and not all admitted patients. In the present study, the control group therefore consisted of a mix of negative patients and patients not tested. However, all patients with COVID-19-like symptoms were tested. Therefore, it is reasonable to assume that the patients not tested for SARS-COV-2 did not have clinical findings of COVID-19.”

2. Below is your response to the previous question on blood culturing frequency. Please include part of this response in the results and discussion.

“The study included patient samples from six tertiary care hospitals. Therefore, it is rather difficult to obtain exact numbers of patients treated during this period. The COVID-19 patients were in total 2240 (27%), of all patients (8262) sampled for blood cultures during the study period. The total number of blood cultured patient during the study period, 8262, is an increase of 1421 (20%), compared to the same period last year (total 6841 patients). Normally we have a yearly 10% increase in numbers of our blood culture bottles at Karolinska without any change in contamination rates. Therefore, it is highly unlikely that the increase in number of contaminated blood cultures is related to more blood cultures being obtained.

In the revised “Materials and methods” section the following information has been added as the Editor suggested (L75-77): Historically, we have a yearly 10% increase in numbers of our blood culture bottles at Karolinska University Laboratory without any significant change in contamination rates.

In the revised “Results” section the following information has been added as the Editor suggested (L144-147): “The COVID-19 group consisted of 2,240/8,262 (27%), of all patients sampled for blood cultures during the study period in 2020. The total number of blood cultured patients during the 2020 study period, 8,262, is an increase of 1,421 (20%), compared to the same period in 2019 (total 6,841 patients).”

3. Additionally in the results, you report that: ”BCs were obtained from 2,240 patients with COVID-19. In 511/2,240 (22.8%) patients, there were two or more episodes during the study period. In the control groups, BCs were obtained from 6022 and 6841 patients in control group-2020 and control group-2019, respectively. In control group-2020, 459 (7.6%) patients had two or more episodes. In control group-2019, 910 (13.3%) patients had two or more episodes.”

The above suggests that the frequency of repeat culturing was higher in the COVID-19 group. I would add that to the discussion.

In the revised “Discussion” section the following information has been added as the Editor suggested (L250-253): There were higher numbers of patients with two or more suspected BSI episodes in the COVID-19 group compared to the two control groups. The underlying reason for this difference might be longer overall hospital stay and/or prolonged ICU stay in patients with COVID-19.

4. Thank you for adding Table 2 and distribution of blood cultures by hospital location. However, the comparison of blood culture contamination rates across locations requires looking at contaminated blood cultures as a proportion of the total blood cultures obtained at that location (and not the total positives). Typically ED contamination rates higher than inpatient locations. You want to assess whether that difference was similar for COVID-19 patients vs controls. Please make changes to Table 2 to reflect that, and add to discussion as appropriate.

We thank the Editor for this very relevant feedback. Table 2 has been revised with the inclusion of contamination rates as proportion of the total BSI episodes as suggested. In addition, the data is discussed as follows: (L287-294) “. In the present study, contamination rates were higher in emergency departments and ICUs compared to other units in the COVID-19 group. The relationship between high stress environments and BC contamination rates has been previously reported [13, 14]. The patient characteristics and the work pace in emergency departments and ICUs differs from the other units. It is reasonable to suggest that the stressful working environment in these two units with well-known risk to be exposed with SARS-CoV-2 might play an important role in higher contamination rates observed in COVID-19 group.”

5. Once again, in the discussion you are comparing your “positivity” rate to bacteremia incidence in published studies. I would clearly state that while you don’t have true incidence in your study, the positivity rate is similar to incidence of bacteremia reported in other studies.

“In the present study the overall blood culture positivity rate was similar in all three groups analyzed. However, the proportion of episodes with clinically relevant growth was significantly lower in in patients with COVID-19 than both control groups. This correlates with previous data on clinical characteristics in COVID-19, where bacteremia was observed in 5.6% of cases and septic shock in 4% of cases”

In the present “Discussion” section the comparison of positivity rate to previous studies has been revised as the editor suggested. The present sections are as follows:

(L241-244): Although in our study the true incidence of bacteremia was not known, the proportion of episodes with clinically relevant growth correlates with previous data on clinical characteristics in COVID-19, where bacteremia was observed in 5.6% of cases [5] and septic shock in 4% of cases [6]. 

(L256-260): The present data differs from the New York study since the overall BC positivity rate in our COVID-19 group did not differ from controls. The underlying reason for this difference is unknown. It is reasonable to assume that the blood culture routines and the characteristics of COVID-19 patients were different. However, both studies showed that clinically relevant growth was lower in patients with COVID-19 than in controls.

---

## [Editor Report · Decision Letter 2]

30 Oct 2020

PONE-D-20-20448R2

Low prevalence of bloodstream infection and high blood culture contamination rates in patients with COVID-19

PLOS ONE

Dear Dr. Özenci,

Thank you for submitting your manuscript to PLOS ONE. After careful consideration, we feel that it has merit but does not fully meet PLOS ONE’s publication criteria as it currently stands. Therefore, we invite you to submit a revised version of the manuscript that addresses the points raised during the review process.

Thank you for the edits thus far. I have one other recommendation with respect to the results text associated with Table 2. The text of the results should be changed to reflect Table 2 edits. Currently it reads as follows:

"When relationship between the frequency of contaminants and the hospital localization was analyzed, both control groups had higher contamination rates in ICU than in emergency departments and other clinics. In contrast, the contamination rates were similar among emergency departments, ICU, and other clinics for patients with COVID-19 (Table 2)." 

Relative to control groups, contamination rates are higher in all locations but more so in ED and ICU locations. It also appears that the contamination rates are not different by location in control group 2020.

We look forward to receiving your revised manuscript.

Kind regards,

Surbhi Leekha

Academic Editor

PLOS ONE

---

## [Author Response · Author response to Decision Letter 2]

2 Nov 2020

We thank the Editor for clarifying the findings presented in Table 2. The differences in contamination rates are important to clarify. A correction has been made to reflect the Editor’s remarks. The revised paragraph commenting Table 2 is as follows:

”When relationship between the frequency of contaminants and the hospital localization was analyzed, control group-2019 had higher contamination rates in ICUs than in emergency departments and other clinics. In contrast, the contamination rates were similar in all hospital locations for control group-2020. In the COVID-19 group, contamination rates were higher in all hospital locations compared to the control groups, but more so in the emergency departments and ICUs (Table 2).”

---

## [Editor Report · Decision Letter 3]

5 Nov 2020

Low prevalence of bloodstream infection and high blood culture contamination rates in patients with COVID-19

PONE-D-20-20448R3

Dear Dr. Özenci,

We’re pleased to inform you that your manuscript has been judged scientifically suitable for publication and will be formally accepted for publication once it meets all outstanding technical requirements.

Kind regards,

Surbhi Leekha

Academic Editor

PLOS ONE
---

## [Editor Report · Acceptance letter]

12 Nov 2020

PONE-D-20-20448R3 

Low prevalence of bloodstream infection and high blood culture contamination rates in patients with COVID-19 

Dear Dr. Özenci:

I'm pleased to inform you that your manuscript has been deemed suitable for publication in PLOS ONE. Congratulations! Your manuscript is now with our production department. 

Kind regards, 

on behalf of

Dr. Surbhi Leekha 

Academic Editor

PLOS ONE